# Analysis of the importance and priority of male and female players in mixed doubles table tennis

Qing Yang[1], Muzi Li[2]*

1 School of Physical Education, Soochow University, Suzhou, Jiangsu, China, 2 Suzhou University of Science and Technology Tianping College Nanjing Campus Public Teaching Department, Nanjing, Jiangsu, China

* lmz603160@163.com

## Abstract

To examine the relative importance and tactical priorities of male and female players in mixed doubles table tennis, this study analysed 35 matches involving world top-50 ranked players. A novel Five-Phase and Four-Mode Model was eveloped and applied. Performance was then evaluated using multiple regression and total decision coefficient method. The results indicate that: (1) Compared to conventional methods, the proposed model demonstrates superior alignment with mixed doubles characteristics in three key dimensions: phase segmentation, assessment indicators, and data acquisition protocols. This methodological advancement significantly enhances the characterization of doubles-specific technical and tactical patterns. (2) Male players demonstrated a higher performance impact (1.3 times that of female players), which is consistent with empirically observed "male-dominant, female-supportive" dynamic in mixed doubles. (3) The "lady-first" strategy is confirmed as an effective match arrangement principle: during service round, the female player should serve first; in receiving scenarios, regardless of the server's identity, the female player should be the primary receiver.

## Introduction

Table tennis is a net sport [1] and a popular official event in the Olympic Games. Competitions generally comprise five events: men's singles, women's singles, men's doubles, women's doubles, and mixed doubles. Mixed doubles is unique as it is the only event where players of opposite genders compete as partners. Players (two pairs) take turns striking the ball, with performance influenced by a range of factors, including technical-tactical skills [2], physical fitness [3,4] and the quality of male-female cooperation [5]. Consequently, even two outstanding singles players may not necessarily form a successful doubles pair [6]. The mixed doubles event debuted at the inaugural World Table Tennis Championships in 1926 and was officially included in the Olympic program at the Tokyo 2020 Games. In recent years, the world table

**Data availability statement:** All relevant data are within the manuscript and its Supporting Information files.

**Funding:** This study was supported by Social Science Foundation of Jiangsu Province (NO. 23TYD006).

**Competing interests:** The authors have declared that no competing interests exist.

tennis landscape has undergone significant generational change, with a new cohort of young players emerging as strong contenders worldwide. Concurrently, scientific research on mixed doubles has accelerated, as scholars investigate its underlying principles from various perspectives and employing relevant methodologies to advance its theoretical foundations [7–9].

Table tennis performance can be assessed through various methodological approaches [10–12], such as three-phase evaluation theory [13], stroke performance relevance model [14], Markov chain model [15,16], expert knowledge model [2], artificial neural network model [17], effectiveness evaluation method [18,19], and comprehensive evaluation method [20]. However, these methods appear more applicable to singles matches. Regarding mixed doubles analysis, Wu et al. proposed technical evaluation criteria for doubles and applied them in international championship preparation, successfully aiding participant selection [21]. Despite new analytical methods being introduced [22,23], the "three-phase evaluation theory" remains predominant in practice research [24–26]. The theory, however, has significant limitations when applied to mixed doubles, including the rationale behind its three-phase division and the applicability of its evaluation criteria. A major drawback is that it treats a pair as a single unit for analysis, thereby failing to assess performance differences between male and female players or between different match sequences (rounds)—factors widely recognized as critically important in mixed doubles [8,22].

Therefore, this study proposes a Five-Phase and Four-Mode Model specifically designed for mixed doubles table tennis and employs this framework for systematic match analysis. The research has two primary objectives: (1) to introduce a total decision coefficient (TDC) based on multiple regression analysis, thereby quantifying the relative contribution of male and female players' performance metrics to match outcomes; and (2) to examine the strategic priority of serving and receiving sequences between genders through a comparative analysis of four distinct stroke orders. Discipline-specific, phase-based analytical methods can more effectively elucidate the core technical and tactical characteristics of a sport. Furthermore, investigating the relative importance and priority of competitive performance contributes to identifying the key determinants of winning in mixed doubles.

## Materials and methods

### Dataset

A total of 35 mixed doubles matches (yielding 70 player-match samples) were selected. All matches involved players ranked within the world top 50, with their opponents also ranked in the top 50 during the match period (2019–2022). All participants were offensive-style players. The sample comprised 30 matches featuring Chinese national team players and 40 matches featuring players from other countries and regions (including Hong Kong China and Taipei China). All match videos were sourced from publicly available television broadcasts or official online platforms. Therefore, written consent was not required for this study. To ensure data reliability, five matches were randomly selected for independent observation and recording by

a second analyst. All observation indicators showed perfect inter-rater agreement (Cohen's $k = 1.00$) [27], conforming their objectivity and the consistency of the coding protocol.

## Performance indicators

**Five-phase and four-mode model in mixed doubles competition.** According to empirical knowledge from previous phased evaluation studies [13,22,28,29], a mixed doubles match is divided into five phases by the Five-Phase and Four-Mode Model: the serving phase (P1), the reserving phase (P2), the third shot phase (P3), the fourth shot phase (P4), and the rallying phase (P5). In this model (Table 1), the rallying phase (P5) comprises the fifth and subsequent shots, not only in the serving round but also in the receiving round. In this paper, the term "stroke" refers to the fundamental hitting technique (e.g., forehand stroke), while "shot" describes its tactical intention and outcome in play (e.g., a winning shot, the third shot).

A table tennis mixed doubles match comprises a total of eight rounds. Denoting the target male and female players as A and B, and their opponents as X and Y, the stroke sequences for all eight rounds are detailed in Table 1. Based on the genders of the server and receiver, these rounds can be categorized into four distinct modes: Male serve to Male receive (MM), Male serve to Female receive (MF), Female serve to Female receive (FF), and Female serve to Male receive (FM).

**Computation of scoring rate, losing rate, and effectiveness.** This study employs scoring rate, losing rate, and effectiveness as performance indicators. Scoring rate reflects the ability to win points in a given phase, losing rate indicates the frequency of conceding points, and effectiveness represents the net benefit per phase. In table tennis, effectiveness is commonly calculated in two ways. Zhang, Liu [18] derived effectiveness through the relationship between scoring rate and usage rate, whereas Tamaki et al. [19] computed it by subtracting the losing rate from the scoring rate. The fundamental distinction is that the former emphasizes the overall gain or loss achieved through the application of a technique (i.e., "total effectiveness"), while the latter evaluates technical efficiency based solely on the point differential attributable to the technique itself, independent of its frequency of use (i.e., "net effectiveness"). To isolate the pure technical performance of male and female players in mixed doubles, this study adopted the computational method proposed by Tamaki et al. [19]. All strokes executed by both players was systematically recorded. Subsequently, the data were aggregated into the proposed five-phase, four-mode analytical framework according to the classification criteria detailed in Table 1. In this system, each stroke results in one of three outcomes: scoring (point won), losing (point conceded), or neutral (rally continues). Let $i$ ($i = 1,2,3,4,5$) denote the phase numbers. For each phase $i$, Let $S_i$ be the number of scoring shots, let $L_i$ be the number of losing shots, and $N_i$ be the number of neutral shots. The performance indicators are then calculated using the following equations.

$$SR_i \ (\text{scoring rate in } i-\text{th phase}) \ = \ S_i \ / \ (S_i + L_i + N_i) \tag{1}$$

**Table 1. Five-Phase and Four-Modes Model in mixed doubles.**

| Round | First shot (P1) | Second shot (P2) | Third shot (P3) | Fourth shot (P4) | Fifth shot and beyond (P5) | Mode | Code |
|---|---|---|---|---|---|---|---|
| 1 | A | X | B | Y | A-X-B-Y | Male serve to Male receive | MM |
| 2 | X | A | Y | B | X-A-Y-B | | |
| 3 | A | Y | B | X | A-Y-B-X | Male serve to Female receive | MF |
| 4 | X | B | Y | A | X-B-Y-A | | |
| 5 | B | Y | A | X | B-Y-A-X | Female serve to Female receive | FF |
| 6 | Y | B | X | A | Y-B-X-A | | |
| 7 | B | X | A | Y | B-X-A-Y | Female serve to Male receive | FM |
| 8 | Y | A | X | B | Y-A-X-B | | |

$$LR_i \ (\text{losing rate in } i-\text{th phase}) \ = \ L_i \ / \ (S_i + L_i + N_i) \tag{2}$$

$$E_i \ (\text{effectiveness in } i-\text{th phase}) = \ SR_i - \ LR_i \tag{3}$$

**Multiple regression and total decision coefficient**

Multiple regression analysis is used to obtain the standard regression coefficients (*SRC*). Taking the effectiveness for male and female players on five phases ($E_{m1}$, $E_{m2}$, $E_{m3}$, $E_{m4}$, $E_{m5}$, $E_{f1}$, $E_{f2}$, $E_{f3}$, $E_{f4}$, $E_{f5}$) as independent variables and *P* as the dependent variable (Equation 4), the regression model for table tennis mixed doubles was established, as shown in Equation 5.

$$P \ (\text{winning probability}) = \ S_{all} \ / \ (S_{all} + L_{all} \ ) \tag{4}$$

$$P = b_0 + b_{01}E_{m1} + b_2E_{m2} + b_3E_{m3} + b_4E_{m4} + b_5E_{m5} + b_6E_{f1} + b_7E_{f2} + b_8E_{f3} + b_9E_{f4} + b_{10}E_{f5} + e \tag{5}$$

In this equation, "$b_0$" is a constant, "$b_1$, $b_2$, $b_3$, $b_4$, $b_5$, $b_6$, $b_7$, $b_8$, $b_9$, $b_{10}$" are pending parameters, and *e* is the error term.

The total decision coefficient (TDC) is calculated as the product of the correlation coefficient between an independent variable and the dependent variable and its standard regression coefficient. The TDC quantifies the percentage of variance in the dependent variable that can be explained by the independent variable, representing its total influence on the dependent variable through all pathways [30]. Thus, the TDC serves as a measure of each variable's relative importance in affecting match outcomes, computed using the following equation:

$$TDC_{Em/fi} = SRC_{Em/fi} \times R_{Em/fiP} \times 100\% \tag{6}$$

In equation 6, $SRC_{Em/fi}$ denotes the standardized regression coefficient of effectiveness(for male or female players) in the multiple regression model, and $R_{Em/fiP}$ represents the correlation coefficient between an independent variable ($E_{mi}$ or $E_{fi}$) and the dependent variable (*P*).

**Statistical analysis**

Statistical analysis were performed to compare the performance between male and female players in mixed doubles and examine the differences across the four modes. The normality of the data for each gender group was assessed using the Shapiro–Wilk test. For comparisons between genders, if the data for both male and female players were normally distributed, an independent samples t-test was applied; Otherwise, the Mann–Whitney U test was used. A one-way ANOVA was employed when data for all four modes were normally distributed; otherwise, the Kruskal–Wallis H test was used. If a significant overall difference was detected ($p < 0.05$), post-hoc pairwise comparisons were conducted using the Bonferroni correction. All analyses were performed at a 95% confidence level, with $p < 0.05$ considered statistically significant. The initial server/receiver selections based on gender and their associated match-winning percentages were analyzed using Pearson's chi-square test. All statistical analyses were conducted using SPSS version 26.0 software for Mac.

## Results

### Multiple regression model and TDC value

Table 2 presents the means, standard deviations, and correlation coefficients of the performance indicators of table tennis players. The Pearson correlation coefficient interval among all independent variables is [−0.221, 0.339], indicating low or

**Table 2. Mean and standard deviation, correlation coefficient and TDC values.**

| | Mean±SD | Correlation coefficient | | | | | | | | | | | TDC % |
| | | $E_{m1}$ | | $E_{m2}$ | $E_{m3}$ | $E_{m4}$ | $E_{m5}$ | $E_{f1}$ | $E_{f2}$ | $E_{f3}$ | $E_{f4}$ | $E_{f5}$ | $P$ |
|---|---|---|---|---|---|---|---|---|---|---|---|---|---|
| $E_{m1}$ | 0.125±0.099 | 1 | 0.201* | 0.151 | 0.004 | 0.140 | 0.035 | 0.248* | −0.054 | 0.076 | 0.299** | 0.416** | 6.5 |
| $E_{m2}$ | 0.102±0.146 | | 1 | 0.115 | 0.124 | −0.114 | 0.187 | −0.071 | −0.024 | −0.093 | 0.167 | 0.273* | 4.7 |
| $E_{m3}$ | 0.073±0.182 | | | 1 | −0.070 | 0.330** | −0.082 | 0.175 | −0.008 | 0.174 | 0.098 | 0.462** | 11.6 |
| $E_{m4}$ | −0.099±0.249 | | | | 1 | 0.165 | 0.011 | 0.204* | 0.208* | −0.054 | 0.099 | 0.413** | 10.8 |
| $E_{m5}$ | −0.151±0.234 | | | | | 1 | 0.057 | 0.339** | −0.080 | 0.177 | 0.161 | 0.595** | 19.5 |
| $E_{f1}$ | 0.105±0.080 | | | | | | 1 | −0.018 | −0.128 | −0.221* | −0.138 | 0.124 | 2.5 |
| $E_{f2}$ | 0.065±0.151 | | | | | | | 1 | 0.176 | 0.143 | 0.166 | 0.518** | 9.0 |
| $E_{f3}$ | −0.026±0.201 | | | | | | | | 1 | 0.280** | 0.225 | 0.338** | 6.6 |
| $E_{f4}$ | −0.111±0.229 | | | | | | | | | 1 | −0.001 | 0.342** | 7.7 |
| $E_{f5}$ | −0.171±0.219 | | | | | | | | | | 1 | 0.502** | 14.0 |
| $P$ | 0.498±0.087 | | | | | | | | | | | 1 | |
| $e$ | | | | | | | | | | | | | 7.0 |

Note: The absence of an asterisk denotes that $P>0.05$, the presence of one asterisk (*) denotes that $P<0.05$, the presence of two asterisks (**) denotes that $P<0.01$.

negligible intercorrelations. The correlations between each independent variable ($E_{mi}$/ $E_{fi}$) and the dependent variable ($P$) is [0.124, 0.595]. With the exception of $R_{Ef1P}$ (0.124), all were statistically significant.

Table 3 presents the the diagnostic results for the regression model. The Durbin-Watson statistic was 1.951, indicating independence of the residuals. An assessment of multicollinearity showed that the variance inflation factor (VIF) for all independent variables ranged from 1.0 to 2.0, confirming the absence of multicollinearity. Normality diagnostics suggested that the residuals approximately followed a normal distribution. The overall regression model was statistically significant ($F=75.369$, $P<0.001$), and an $R^2$ of 0.927, indicating an excellent model fit.

Based on Equation 6, the TDC values of each performance indicator were calculated and presented in Table 2. The results indicate significant differences in TDC between male and female players. Specifically, the values were as follows: for $TDC_1$, 6.5% (male) versus 2.5% (female); for $TDC_2$, 4.7% versus 9.0%; for $TDC_3$, 11.6% versus 6.6%; for $TDC_4$, 10.8%

**Table 3. Results of multiple regression model.**

| | B | β | t | p | 95%CI | VIF | $R^2$ | F | F(sig) | Durbin-Watson |
|---|---|---|---|---|---|---|---|---|---|---|
| Constants | 0.490 | | 61.500 | 0.000 | 0.474, 0.506 | 1.249 | 0.927 | 75.369 | 0.000** | 1.951 |
| $E_{m1}$ | 0.138 | 0.157 | 4.006 | 0.000 | 0.069, 0.208 | 1.237 | | | | |
| $E_{m2}$ | 0.102 | 0.171 | 4.376 | 0.000 | 0.055, 0.148 | 1.237 | | | | |
| $E_{m3}$ | 0.119 | 0.25 | 6.418 | 0.000 | 0.082, 0.157 | 1.198 | | | | |
| $E_{m4}$ | 0.091 | 0.261 | 6.795 | 0.000 | 0.064, 0.118 | 1.468 | | | | |
| $E_{m5}$ | 0.122 | 0.327 | 7.702 | 0.000 | 0.090, 0.153 | 1.169 | | | | |
| $E_{f1}$ | 0.218 | 0.201 | 5.306 | 0.000 | 0.136, 0.300 | 1.294 | | | | |
| $E_{f2}$ | 0.101 | 0.174 | 4.367 | 0.000 | 0.054, 0.147 | 1.341 | | | | |
| $E_{f3}$ | 0.085 | 0.195 | 4.797 | 0.000 | 0.049, 0.120 | 1.271 | | | | |
| $E_{f4}$ | 0.085 | 0.224 | 5.656 | 0.000 | 0.055, 0.115 | 1.302 | | | | |
| $E_{f5}$ | 0.111 | 0.278 | 6.956 | 0.000 | 0.079, 0.142 | 1.249 | | | | |

Note: B represents Unstandardized coefficients; β represents Standardization coefficients. The absence of an asterisk denotes that $P>0.05$, the presence of one asterisk (*) denotes that $P<0.05$, the presence of two asterisks (**) denotes that $P<0.01$.

versus 7.7%; and for $TDC_5$, 19.5% versus 14.0%. Consequently, the TDC rankings for male players were $TDC_5 > TDC_3 > TDC_4 > TDC_1 > TDC_2$, whereas for female players they were $TDC_5 > TDC_2 > TDC_4 > TDC_3 > TDC_{1。}$

## Performance difference between male and female players in mixed doubles

Fig 1 displays the distribution of the scoring rate, losing rate, and effectiveness for male and female players. A significant difference in scoring rate was observed between genders in P3 ($t = 3.486$, $p < 0.01$; see Fig 1a). In contrast, no significant gender difference was found in losing rate (Fig 1b). For effectiveness, a significant gender difference was also identified in P3 ($t = 3.027$, $p < 0.01$; see Fig 1c).

## Performance difference between four modes in mixed doubles

Fig 2 presents the distribution of scoring rate, losing rate, and effectiveness across the four modes, highlighting significant differences primarily concentrated in P3 and P4. A significant difference in scoring rate among the four modes was identified in P3 ($H = 17.102$, $p < 0.01$; see Fig 2a). Post hoc analysis revealed that the scoring rates for both FF and FM modes were significantly higher than that of the MM mode (both $p < 0.01$). A significant difference in losing rate among the four modes was found in the P4 ($H = 10.977$, $p < 0.05$; see Fig 2b). Post hoc tests indicated that the losing rate for FM mode was significantly higher than that for the MM mode ($p < 0.05$). Significant differences in effectiveness among the four modes were identified in both P3 ($F = 5.153$, $p < 0.01$) and P4 ($H = 12.143$, $p < 0.01$) (see Fig 2c). Post hoc comparisons showed that in P3, the effectiveness value for FF and FM modes were both significantly greater than that for the MM mode (both $p < 0.01$). In P4, the effectiveness for the FM mode was significantly lower than that for the MM mode ($p < 0.01$).

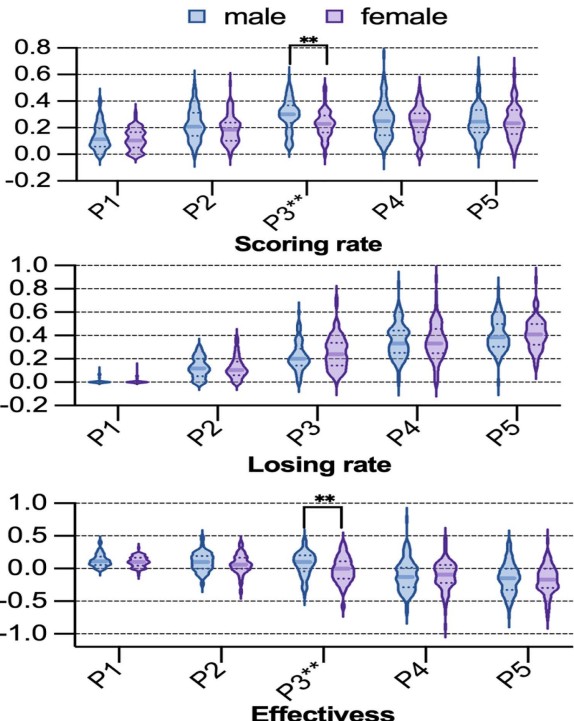

**Fig 1. Scoring rate, losing rate, and effectiveness between male and female players.** Note: The absence of an asterisk denotes that $P > 0.05$, the presence of one asterisk (*) denotes that $P < 0.05$, and the presence of two asterisks (**) denotes that $P < 0.01$.

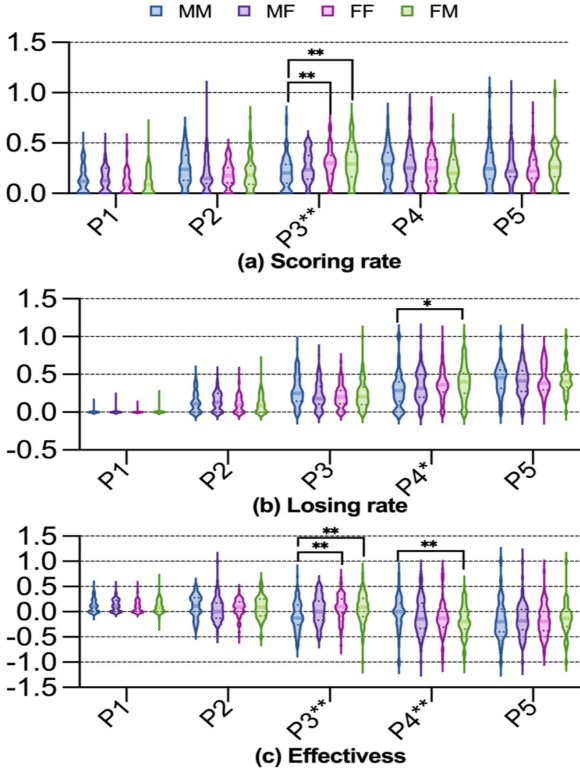

**Fig. 2. Scoring rate, losing rate, and effectiveness between four modes.** Note: The absence of an asterisk denotes that P > 0.05, the presence of one asterisk (*) denotes that P < 0.05, and the presence of two asterisks (**) denotes that P < 0.01.

## Discussion

### Phase-division rationale in mixed doubles

**Theoretical foundations for Five-Phase and Four-Mode Model construction.** This study is the first to propose a Five-Phase and Four-Mode Model specifically for mixed doubles, grounded in three key rationales: (1) The initial four shots in mixed doubles are decisive for match outcomes. In table tennis, most rallies conclude within the first four shots [14], a pattern particularly pronounced in mixed doubles. As illustrated in Fig 3, scoring in mixed doubles is heavily concentrated in these initial shots, with point percentages of 12.2%, 20.6%, 23.5%, and 16.8% for the first through fourth shots, respectively, cumulatively accounting for 73.1% of all points. Consequently, these initial four shots warrant focused attention in performance analysis. (2) Significant differences exist between male and female players in stroke techniques and quality [6,24], resulting in weaker continuity during the initial four shots compared to singles or other double events. Consequently, segmenting mixed doubles matches combining the first and third shots into a unified "serve and attack" phase, and the second and fourth into a "receive and attack" phase—as is common in singles analysis—is inappropriate. Instead, each of the first four shots should be analyzed as a distinct tactical unit rather than being grouped into broader phases for macro-level analysis. Furthermore, the key advantage of this fine-grained segmentation lies in its analytical flexibility: it retains the capability to analyze individual performance data for all four players while simultaneously enabling integrated, pair-level analysis through strategic data combination as needed.

(3) Beyond the fourth stroke, scoring rates decline markedly, as most rallies transition to topspin-driven exchanges. Consequently, grouping all shots from the fifth onward into a unified "rally phase" aligns with the inherent dynamics of mixed doubles.

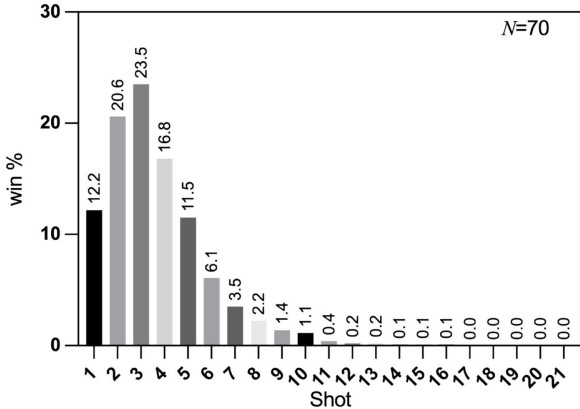

**Fig. 3. Percentage of points scored per shot in mixed doubles matches.** Note: The percentage of points scored on each shot is calculated as the percentage of points scored on that shot out of all points scored in the match.

### Comparison with conventional methods.

(1) Differences in evaluation metrics

Most existing analytical approaches for mixed doubles table tennis are fundamentally derived from the "three-phase evaluation method", which this study collectively refers to as the "conventional method". This traditional approach primarily employs two metrics: scoring rate (SR) and usage rate (UR). In contrast, the proposed Five-Phase and Four-Mode Model introduces three refined indicators: scoring rate (SR), loss rate (LR), and technical effectiveness (E).

Key Differences in Evaluation Metrics: (a) Both methods SR to assess offensive effectiveness. However, conventional SR calculations exclude neutral shots (i.e., rallies that neither score nor lose a point), resulting in less precise measurements compared to those generated by the proposed model. (b) UR reflects the distribution of scoring and losing shots across phases, calculated as: $UR = (Points_{won} + Points_{lost})_{phase} / Total\ match\ points \times 100\%$ [13]. In the proposed model, UR becomes redundant because the first four shots are analyzed as distinct tactical units, thereby eliminating the need for a metric that quantifies their distribution across aggregated phases. (c) LR quantifies defensive vulnerability in each phase, while E integrates SR and LR to evaluate net performance. Consequently, the proposed model generates more comprehensive and precise technical-tactical data. This enables not only the identification of players' strengths and weaknesses in mixed doubles, but also root-cause diagnostics through the distinct scoring and loss rate metrics.

(2) Differences in data collection

The conventional method adopts the final shot executed by the observed player(s) as the observational unit, exclusively recording shots that directly determine point outcomes (i.e., those that win or lose the rally) [14]. Despite its operational simplicity, this method suffers from critical limitations in data comprehensiveness. By design, it restricts performance metrics to the observed pair, thereby necessitating separate recollection of opponent data. Moreover, the aggregated team-level statistics fail to disentangle gender-specific contributions, thereby obscuring individual performance disparities between male and female players. In contrast to conventional approaches, the proposed model eliminates data asymmetry through comprehensive, per-shot tracking of all four players. This systematic data acquisition enables actionable insights by reconstructing paired interaction dynamics specific to mixed doubles.

**Four modes and four groups.** The four modes in present study correspond to four distinct stroke sequences. Zhou and Zhang [7] previously classified mixed doubles interaction into four groups based on the gender matchup in the final two shots of a rally: "male versus male ($P_{m-m}$), male versus female ($P_{m-f}$), female versus male ($P_{f-m}$), and female versus

female (Pf$_{-f}$) ". In contrast to the present model, their classification was derived by aggregating the outcomes and the player genders of the last two shots, irrespective of the competition order (i.e., service sequence) or the specific shot number within the rally. For example, if a male player scores on a shot and the opposing female player loses the point on the subsequent shot, the rally is categorized as P$_{m-f}$. A fundamental limitation of this grouping method is that each group can theoretically occur in any stroke sequence (order) and at any point within the prescribed rules of play. Consequently, based on their finding that "P$_{m-f}$ present larger direct path coefficients and the total determined coefficient", their model would logically suggest initiating play with the male player serving first, rather than the female player. The essential reason is that their grouping method cannot pinpoint at which specific stroke the P$_{m-f}$ interaction exerts its greatest influence. Consequently, their conclusion was based on a hypothetical extrapolation from professional experience—specifically, that the importance of P$_{m-f}$ lies in the third and fourth shots, rather than in the first two (serve and receive). In contrast, the present study is grounded in a statistical analysis of each shot within the four defined modes (stroke sequences). This approach thereby enables an accurate and robust conclusion regarding stroke ordesr in mixed doubles.

## Importance of gender-dependent performance in mixed doubles

TDC quantifies the relative contribution of various technical indicators (*Em$_1$*–*Em5*, *Ef$_1$*–*Ef$_5$*) to the probability of winning a match, representing the importance of each technique for male and female players in determining match outcomes. The results indicate that the total TDC is 53% for male players and 39.7% for female players. This implies that in mixed doubles, 53% of the match outcome variance is attributable to male performance, while 39.7% is attributable to female performance, male players' performance is almost 1.3 times more important than female players.

In addition, the most important effectiveness for both male and female players occurred in P5 (19.5%, 14%). Zhou and Zhang [7] concluded that male players outperformed female players in the first four shots. Similarly, Peng [24] found that male players demonstrated superior performance in both the "attack after serving" and "attack after receiving" phases, indicating a pronounced gender-based advantage. These prior studies statistically compared the performance of male and female players within specific match contexts and extrapolated that whichever gender held a technical advantage in those contexts would be more decisive for the match outcome. In contrast, the present study diverges by examined the overall influence of male versus female players performance on the match outcome as an integrated system. It further quantifies this relative importance by employing multiple regression analysis to calculate specific Total Decision Coefficient (TDC) values..

## Priority of gender-dependent performance in mixed doubles

The results indicate that male players demonstrated superior skills to female players specifically in P3 of mixed doubles (Fig 2a), with no significant gender difference observed in other phases. Given that the effectiveness disparity originated from scoring rate rather than losing rate, the performance gap is primarily attributable to the scoring advantage conferred by male players' third-stroke skills. While previous studies established that gender-based differences predominantly occur within the initial four shots [7,24], the present study precisely identifies the exact location of this difference.

The significant differences among the four modes were concentrated P3 and P4. (1) In P3, both he scoring rate and effectiveness for the FF and FM modes were significantly higher than those for the MM mode. A key commonality between the FF and FM modes in P3 is that the striking player is male (Player A or X), with both modes achieving the highest scoring rate (approximately 0.30), In contrast, in the MM mode, the striking player is female (Player B or Y), resulting in the lowest scoring rate (0.203). These findings further confirm that the scoring advantage of male players is significantly greater than that of female players in P3. (2) In P4, the significantly higher losing rate for the FM mode compared to the MM mode may be attributed to the FM mode's highest scoring rate in the preceding stroke (P3). A scoring advantage in one shot often creates a tactical disadvantage in the subsequent shot. It is noteworthy that while the scoring rates for

both the FF and FM modes were significantly higher than that of the MM mode, a significant difference in losing rate was observed only between the FM and MM modes. This is primarily because the player executing the fourth shot in the FF mode is the male (Player X or A), whereas in the FM mode, it is the female (Player Y or B). Evidently, when confronting an advantage stroke delivered by an opposing male player in the third shot, male player demonstrate better performance in the subsequent fourth shot compared to female player.

In summary, male player has an absolute technical advantage in modes where they execute the third shot (FF and FM) in mixed doubles. To counteract this, a feasible tactical response for the opposing team is to position their male player to strike the fourth shot.. Consequently, a pre-match strategic arrangement, which may be termed a "female-first" strategy, can be formulated as follows: during service rounds, the female player serves first to ensure the male partner strikes the third shot; during receiving rounds, regardless of the server's identity, the female player receives first to enable the male partner to strike the fourth shot.

## Limitations

First, this study conducted a holistic analysis on elite players ranked within the world's top 50; its findings are therefore relative and may not be fully generalizable to other player populations, such as youth or amateur athletes. Moreover, practical training should account for individual differences in playing style and personality traits among elite players. Second, the analysis is based on player performance under current mixed doubles competition rules. As rules evolve and techniques and tactics develop, further longitudinal tracking studies are warranted.

## Conclusion

This study conducted a performance analysis to investigate the relative importance and tactical priority of male and female players in mixed doubles table tennis. A novel Five-Phase and Four-Mode Model was proposed, which demonstrates superior alignment with mixed doubles characteristics across three dimensions: phase segmentation, assessment indicators, and data acquisition protocols. Male players exhibited a performance impact 1.3 times (or 13.3%) greater than female counterparts, consistent with the empirically observed "male-dominant, female-supportive" dynamic in mixed doubles. The "lady-first" strategy is validated as effective for match arrangements: during the service round, the female player should serve first; in receiving scenarios, regardless of server's identity, the female player should be the primary receiver.

## Supporting information

**S1 File. Raw data.**
(ZIP)

## Author contributions

**Conceptualization:** Muzi Li.

**Data curation:** Qing Yang.

**Formal analysis:** Qing Yang, Muzi Li.

**Funding acquisition:** Qing Yang, Muzi Li.

**Methodology:** Qing Yang.

**Visualization:** Muzi Li.

**Writing – original draft:** Qing Yang, Muzi Li.

**Writing – review & editing:** Qing Yang, Muzi Li.

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
