## [Decision Letter · Decision Letter 0]

19 Sep 2025

Dear Dr. Li,

Thank you for submitting your manuscript to PLOS ONE. After careful consideration, we feel that it has merit but does not fully meet PLOS ONE’s publication criteria as it currently stands. Therefore, we invite you to submit a revised version of the manuscript that addresses the points raised during the review process.

We look forward to receiving your revised manuscript.

Kind regards,

Alejandro Torrado Pacheco, PhD

Associate Editor

PLOS One

Journal Requirements:

“This study was supported by Social Science Foundation of Jiangsu Province (NO. 23TYD006).”

3. We note that your Data Availability Statement is currently as follows:

“All relevant data are within the manuscript and its Supporting Information files.”

Additional Editor Comments:

The manuscript has been evaluated by two reviewers, and their comments are available below. The reviewers raised concerns with the theoretical justification for the conceptual model used, and requested that the study be better framed within existing analysis frameworks.

Could you please carefully address the comments raised?

Reviewer's Responses to Questions

**Comments to the Author**

1. Is the manuscript technically sound, and do the data support the conclusions?

Reviewer #1: Yes

Reviewer #2: Yes

2. Has the statistical analysis been performed appropriately and rigorously?

Reviewer #1: Yes

Reviewer #2: Yes

3. Have the authors made all data underlying the findings in their manuscript fully available?

Reviewer #1: Yes

Reviewer #2: Yes

4. Is the manuscript presented in an intelligible fashion and written in standard English?

Reviewer #1: Yes

Reviewer #2: Yes

Reviewer #1: Overall Evaluation:

This paper presents a study on mixed doubles table tennis analysis using a novel Five-Phase Four-Mode Model. While the topic is relevant and the model has potential, several issues need to be addressed to improve clarity, accuracy, and scholarly rigor. Below are specific comments and recommendations for revision.

Major Comments:

Terminology Consistency:

The term "ping pong" in the abstract should be replaced with "table tennis" to maintain academic consistency and avoid confusion with sandpaper table tennis.

The interchangeable use of "players" and "athlete" throughout the paper should be standardized as well as "shot" and "stroke". "Players" and "stroke" are recommended for consistency.

Conceptual Clarity of the Five-Phase Four-Mode Model:

The justification for the "Five phase" is unclear. The literature cited (line 117) only supports three phases, not five. The authors must provide a stronger theoretical or empirical basis for defining five phases.

Defining each of the first four shots as separate phases is problematic. This approach conflates the concepts of "shot" and "phase," which have been distinctly defined in previous research. The authors should either revise this terminology or provide a compelling rationale for redefining these terms.

The discussion section (lines 266-272) claims that the weak continuity in mixed doubles justifies treating each shot as a phase. However, this ignores the tactical coordination between teammates across shots (e.g., serve and third-shot attack). The authors should revise this section to align with established tactical analysis frameworks.

Methodological Details:

The consistency metrics mentioned (lines 108-111) need clarification. Specifically, which indicators (e.g., number of shots, technical actions) achieved a consistency value of 1? This must be explicitly stated.

Data Presentation and Analysis:

Table 2 requires layout optimization to enhance readability.

Statistical errors must be corrected:

Line 221: The TDC1 values for male and female players should be stated as "6.5% and 2.5%, respectively."

Lines 225-226: The "TDC order" values contain significant errors and require revision.

Citation formatting:

Line 261: "(14)" should be replaced with "[14]."

Line 266: "(6, 24)" should be replaced with "[6, 24]."

Minor Comments:

Introduction: Remove "singles" from "table tennis singles competition" (line 54) to avoid redundancy with later uses of "singles."

Recommendation:

Minor revisions are required to address the issues above. The authors should thoroughly revise the terminology, clarify the conceptual framework of the Five-Phase Four-Mode Model, which is the important core in this manuscript, correct statistical errors, and improve the presentation of tables and citations.

Reviewer #2: Dear Editor and Authors,

I am grateful for the opportunity to review a paper in your prestigious journal. Paper entitled: "Analysis of the importance and priority of male and female players in mixed doubles." table tennis" is a research paper that contributes to theoretical consideration as well as practical application.

The authors of the paper relied on acceptable theoretical concepts in the introductory part and based on the aim of the paper, as well as on two valid approaches for calculating the efficiency of shots in table tennis (Zhang Liu and Tamaki et al), which are referred to in the part of the subtitle of the paper Computation of SR, LR, and E. I suggest that the mentioned subtitle be redefined, because the authors state abbreviations that do not appear in the text. The authors clearly define two goals that rely on a clear theoretical fashion.

In the Material and methods part, they clearly define the number of observed matches (number of mixed doubles players) of elite rank and validate the objectivity of the observation based on the value of 1 according to Cohen's kappa values (values greater than 0.8). A clear statistical procedure is given, on the basis of which the results are clearly presented in textual and tabular as well as graphically.

In the Discussion part, they clearly rely on the theoretical basis for the application of a new approach in analysis based on five phases and four modalities. It is important that the authors also mention the limitation of the study, which indicates the applicability of the model that could be tested for other competitive or age levels, thereby additionally opening a new research question. In the conclusion, a clear benefit of the proposed model is given, which shows a significant compliance with the characteristics of mixed doubles in table tennis and confirms the already known fact about the efficiency of male table tennis players over female ones.

I suggest the editor to fully accept the work for publication with a minor suggestion about the abbreviations in the subtitle that should be redesigned.

Kind regards,

**Do you want your identity to be public for this peer review?** For information about this choice, including consent withdrawal, please see our Privacy Policy

Reviewer #1: No

Reviewer #2: No

---

## [Author Response · Author response to Decision Letter 1]

2 Nov 2025

Dear reviewers

Thank you very much for your valuable comments and suggestions. We appreciate and are greatly motivated by the kind comments that recognize the potential of our work. We have done our best to address all the concerns raised and revised the paper accordingly.

We have updated the cover letter, supplemented the raw data, and provided point-by-point responses to the reviewers' comments. We hope that this revised version addresses all the concerns of the reviewers. Due to the large amount of content in this revision, if you approve our revision, we will invite native English speakers to polish the full text. Your assistance in reviewing this paper is highly appreciated.

Yours sincerely,

Qing Yang, Mu-zi Li.

---

## [Decision Letter · Decision Letter 1]

22 Dec 2025

Dear Dr. Li,

plosone@plos.org . A letter that responds to each point raised by the academic editor and reviewer(s). You should upload this letter as a separate file labeled 'Response to Reviewers'.A marked-up copy of your manuscript that highlights changes made to the original version. You should upload this as a separate file labeled 'Revised Manuscript with Track Changes'.An unmarked version of your revised paper without tracked changes. You should upload this as a separate file labeled 'Manuscript'.

We look forward to receiving your revised manuscript.

Kind regards,

Jennifer Tucker, PhD

Staff Editor

PLOS One

Journal Requirements:

Reviewers' comments:

Reviewer's Responses to Questions

**Comments to the Author**

Reviewer #3: (No Response)

Reviewer #4: (No Response)

2. Is the manuscript technically sound, and do the data support the conclusions?

Reviewer #3: (No Response)

Reviewer #4: Yes

3. Has the statistical analysis been performed appropriately and rigorously?

Reviewer #3: (No Response)

Reviewer #4: Yes

4. Have the authors made all data underlying the findings in their manuscript fully available?

Reviewer #3: (No Response)

Reviewer #4: Yes

5. Is the manuscript presented in an intelligible fashion and written in standard English?

Reviewer #3: (No Response)

Reviewer #4: Yes

Reviewer #3: (No Response)

Reviewer #4: The revised manuscript demonstrates consistent statistical reporting and clear terminology, reflecting effective integration of previous reviewer feedback. The analytical framework and quantitative methods are clearly described and appropriately applied to mixed doubles table tennis. Only minor editorial and stylistic refinements are recommended to further improve clarity and readability.

**Do you want your identity to be public for this peer review?** For information about this choice, including consent withdrawal, please see our Privacy Policy

Reviewer #3: No

Reviewer #4: No

---

## [Author Response · Author response to Decision Letter 2]

28 Dec 2025

Dear reviewers

Thank you very much for your valuable comments and suggestions. We appreciate and are greatly motivated by the kind comments that recognize the potential of our work. We have done our best to address all the concerns raised and revised the paper accordingly.

We have updated the cover letter, supplemented the raw data, and provided point-by-point responses to the reviewers' comments. We hope that this revised version addresses all the concerns of the reviewers. Due to the large amount of content in this revision, if you approve our revision, we will invite native English speakers to polish the full text. Your assistance in reviewing this paper is highly appreciated.

Yours sincerely,

Qing Yang, Mu-zi Li.

---

## [Decision Letter · Decision Letter 2]

19 Jan 2026

Analysis of the importance and priority of male and female players in mixed doubles table tennis

PONE-D-25-34911R2

Dear Dr. Li,

We’re pleased to inform you that your manuscript has been judged scientifically suitable for publication and will be formally accepted for publication once it meets all outstanding technical requirements.

Kind regards,

Annesha Sil, Ph.D.

Staff Editor

PLOS One

Additional Editor Comments (optional):

Reviewers' comments:

Reviewer's Responses to Questions

**Comments to the Author**

Reviewer #3: (No Response)

Reviewer #4: All comments have been addressed

2. Is the manuscript technically sound, and do the data support the conclusions?

Reviewer #3: (No Response)

Reviewer #4: (No Response)

3. Has the statistical analysis been performed appropriately and rigorously?

Reviewer #3: (No Response)

Reviewer #4: (No Response)

4. Have the authors made all data underlying the findings in their manuscript fully available?

Reviewer #3: (No Response)

Reviewer #4: (No Response)

5. Is the manuscript presented in an intelligible fashion and written in standard English?

Reviewer #3: (No Response)

Reviewer #4: (No Response)

Reviewer #3: (No Response)

Reviewer #4: (No Response)

**Do you want your identity to be public for this peer review?** For information about this choice, including consent withdrawal, please see our Privacy Policy

Reviewer #3: No

Reviewer #4: No

---

## [Editor Report · Acceptance letter]

19 Sep 2025

PONE-D-25-34911R2

PLOS One

Dear Dr. Li,

I'm pleased to inform you that your manuscript has been deemed suitable for publication in PLOS One. Congratulations! Your manuscript is now being handed over to our production team.

Kind regards,

on behalf of

Dr Annesha Sil

Staff Editor

PLOS One